# Development of Policy-Relevant Indicators for Injury Prevention in British Columbia by the Key Decision-Makers

**DOI:** 10.3390/ijerph182211837

**Published:** 2021-11-11

**Authors:** Megan Oakey, David C. Evans, Tobin T. Copley, Mojgan Karbakhsh, Diana Samarakkody, Jeff R. Brubacher, Samantha Pawer, Alex Zheng, Fahra Rajabali, Murray Fyfe, Ian Pike

**Affiliations:** 1BC Injury Research and Prevention Unit, BC Children’s Hospital Research Institute, Vancouver, BC V6H 3V4, Canada; Mojgan.Karbakhsh@bcchr.ca (M.K.); Dianacsamkody@gmail.com (D.S.); samantha.pawer@bcchr.ca (S.P.); Alex.Zheng@bcchr.ca (A.Z.); frajabali@bcchr.ca (F.R.); Ipike@bcchr.ca (I.P.); 2BC Centre for Disease Control, Provincial Health Services Authority, Vancouver, BC V5Z 4R4, Canada; 3Trauma Services BC, 1770 West 7th Ave., Vancouver, BC V5Z 1M9, Canada; David.Evans@vch.ca; 4Fraser Health Authority, 13450 102 Ave., Surrey, BC V3T 5X3, Canada; Tobin.Copley@fraserhealth.ca; 5Department of Emergency Medicine, Faculty of Medicine, The University of British Columbia, Vancouver, BC V5Z 1M9, Canada; Jeff.Brubacher@ubc.ca; 6Vancouver Island Coastal Health Authority, 430-1900 Richmond Ave., Victoria, BC V8R 4R2, Canada; Murray.Fyfe@viha.ca; 7Department of Pediatrics, The University of British Columbia, Vancouver, BC V6H 3V4, Canada

**Keywords:** injury, indicators, surveillance, policy

## Abstract

Indicators can help decision-makers evaluate interventions in a complex, multi-sectoral injury system. We aimed to create indicators for road safety, seniors falls, and ‘all-injuries’ to inform and evaluate injury prevention initiatives in British Columbia, Canada. The indicator development process involved a five-stage mixed methodology approach, including an environmental scan of existing indicators, generating expert consensus, selection of decision-makers and conducting a survey, selection of final indicators, and specification of indicators. An Indicator Reference Group (IRG) reviewed the list of indicators retrieved in the environmental scan and selected candidate indicators through expert consensus based on importance, modifiability, acceptance, and practicality. Key decision-makers (*n* = 561) were invited to rank each indicator in terms of importance and actionability (online survey). The IRG applied inclusion criteria and thresholds to survey responses from decision-makers, which resulted in the selection of 47 road safety, 18 seniors falls, and 33 all-injury indicators. After grouping “like” indicators, a final list of 23 road safety, 8 seniors falls, and 13 all-injury indicators were specified. By considering both decision-maker ranking and expert opinion, we anticipate improved injury system performance through advocacy, accountability, and evidence-based resource allocation in priority areas. Our indicators will inform a data management framework for whole-system reporting to drive policy and funding for provincial injury prevention improvement.

## 1. Introduction

Injuries continue to be an important cause of death and disability in developed as well as developing nations [1]. In 2017, 521 million people worldwide sustained non-fatal injuries and 4.5 million died from injuries, with the main causes of death being road traffic injuries, self-harm, and falls [2]. In Canada, injuries are also a public health concern and the leading cause of death among those aged 1 to 44 years [3]. In 2018, 4.6 million Canadians visited emergency departments for injuries, leading to CAD 20.4 billion in direct health-care costs and CAD 29.4 billion in total economic costs to society [4]. In British Columbia (B.C.), injuries are responsible for over 2500 deaths, 9000 permanent disabilities, 38,000 hospitalizations, and 508,000 emergency department visits each year. Injuries cost the B.C. economy CAD 3.7 billion in 2010 (equivalent to CAD 4.5 billion in 2019), including CAD 2.6 billion in direct costs to the B.C. healthcare system. Unintentional and intentional injuries accounted for 84% (CAD 3.8 billion) and 14% (CAD 0.65 billion) of the total, respectively [5,6,7]. Beyond years of life lost and lived with disability, injuries result in insurmountable consequences for families, communities, the health care system, and society at large [7]. In order to address this staggering burden, the B.C. Ministry of Health prioritized injury prevention as one of the seven visionary goals in its 2017 policy framework: BC’s Guiding Framework for Public Health [8].

Within the past decades, numerous safety promotion and injury prevention efforts have been in effect in B.C. These include, among others, the institution of the B.C. Injury Research and Prevention Unit (BCIRPU), the development of the Canadian Falls Prevention Curriculum [9], Preventable [10,11,12], Road Safety Strategy 2015 and Beyond [13], Concussion Awareness Training Tool (CATT) [14,15], and Active & Safe Central [16]. One of the essential components required for successful and sustainable implementation of public health interventions includes the communication of accurate, timely, and concise information to decision-makers and other key stakeholders [17]. Numerous translational research efforts in the field of injury prevention have scrutinized the “research-to practice” gap—i.e., exploring barriers to the widespread adoption of evidence-based interventions [18,19]. However, an emerging issue of concern in the “practice-to-research” gap is the relevance of research to the needs, as recognized by decision-makers and community stakeholders [20]. The “science push” driven by the emphasis on evidence-based decision-making tends to augment a unidirectional flow of information or the “pipeline fallacy”, which can inhibit contribution by practitioners and decision-makers in program development and evaluation [21].

In B.C., multiple efforts have been made to integrate ‘rigor and ‘reality’ [22], through setting priorities on research needs relevant to decision-makers [23], as well as quantifying the economic costs and societal burden associated with injuries [24]. Although information is available regarding injury statistics, monitoring, and surveillance programs, it might not always be available in a form that allows decision-makers to consider and implement policies. A participatory simulation modelling in Australian real-world policy settings showed that decision-makers preferred reports and statistics from government and non-government organizations over journal articles for informing decisions, as they tended to be free to access and were formulated in a more accessible language [25]. Providing injury statistics in a form that policymakers appreciate and accept, and that can be effectively used to inform policy (considering time and resource constraints) is a major undertaking. This requires the selection of information that is relevant to the priority areas, followed by translation of this information into a consistent and coherent conceptual and operational framework.

Indicators can play an important role in transforming data into relevant information that decision-makers can use. Indicators can be defined as observable metrics (e.g., percentage, rate, number) which can be used to measure a risk or protective factor or a community construct with a theoretical or empirical relationship to a risk or protective factor or outcome [26]. Indicators are, in fact, summary measures that can shed light on a situation which is not obvious when considered by itself [27]. For instance, disaggregated mortality and morbidity indicators across demographic, geographic, and socio-economic dimensions can reveal inequities and highlight priority areas for public health intervention. The disparities in death rate and life expectancy by city neighbourhood and occupation has prompted arguments toward a more egalitarian society and has elicited research to explore causal pathways from as early as the nineteenth century [28,29], as it still does today [30]. Locally derived indicators can also be used for capacity-building and informing decision-making bodies, enabling them to play a more dynamic role in health assessment and subsequent interventions [31]. An injury indicator is “a summary measure which denotes or reflects, directly or indirectly, variations and trends in injuries, or injury-related or injury control-related phenomenon” [32,33]. There has been a renewed international interest in using injury indicators and increasing recognition by decision-makers of the need to use objective landmarks to justify and inform resource allocation, and to evaluate interventions [34,35,36,37].This is in parallel with the recognition that substantial improvement in decreasing the injury burden can only be expected through a ‘systemic approach’ to injury prevention, warranting sustained changes at the societal level targeting population-level injury indicators [38].Using indicators for monitoring progress in the field of injury prevention is of special importance, as the systems in which injury prevention programs are implemented are complex, multi-sectoral, and multi-dimensional [36].

Most injury indicators currently in use in B.C.—similar to indicators in many other jurisdictions [33,34,37,39,40]—focus on outcome measures such as mortality rates, emergency department visits, and hospitalizations [8]. While important, these outcome indicators do not provide a whole-of-system perspective that can accurately inform policy, practice, and resource allocation. Indicators to support injury prevention initiatives need to extend beyond morbidity and mortality outcome measures and include contributory factors in order to gauge the changes in the system required for reducing the burden of injury [36].

In B.C., there was an unmet need for policy-relevant injury indicators to inform decision-makers regarding priority areas and actions that required their attention and management. To this end, the objective of this paper is to identify and define a suite of injury indicators to advance and guide the monitoring and evaluation of injury prevention initiatives in B.C. The research question was: ‘which injury indicators are sufficiently relevant and applicable to the B.C. injury system and reflect the expert opinions as well as decision-makers’ choices?’ These indicators will be used to recommend a data management framework for policy-relevant whole-of-system reporting.

## 2. Methods

### 2.1. Context

The B.C. Injury Prevention Committee (BCIPC) holds the provincial mandate to provide guidance and recommendations on injury prevention policy and practice to the Ministry of Health and the Provincial Health Officer as well as to lead the implementation of various injury reduction initiatives. The mandate for injury surveillance in B.C. is jointly held by the BCIRPU, the B.C. Observatory for Population & Public Health (Observatory), and the BC Ministry of Health (Health Sector Information, Analysis and Reporting).

### 2.2. Priority Setting Process

For determining provincial injury prevention priorities, the BCIPC undertook a rigorous mixed-method three-round modified Delphi approach adapted from the methods described by Lindsay et al. [41] and successfully implemented by Pike et al., in a Canadian context [42,43]. In round 1 of the modified Delphi, a prioritization matrix was used to rate major mechanisms of injury. In round 2, a prioritization matrix was again used, this time to rate sub-mechanisms of injury (43 total). In round 3, the top 10 sub-mechanisms of injury were compared to each other using pairwise comparison, soliciting a vote from each BCIPC member as to which of each pairing should be given higher priority. This step was implemented to validate and adjust the rankings of the top 10 injury sub-mechanisms and to allow for a “sober second thought” on the stage 2 results. This process resulted in the identification of three priorities: (1) seniors falls and fall-related injuries, (2) transport-related injuries (young drivers, pedestrians, cyclists, motor vehicle occupants), and (3) youth suicide and self-harm. Recommendations for action were then developed for road safety and seniors falls by conducting large evidence syntheses with a subsequent iterative expert review to consensus [23,44]. These two priorities further informed the indicator development process, in addition to a third category of ‘all-injury’ indicators [37,45,46] to enable inclusiveness and whole-of-system surveillance. Indicators for youth suicide and self-harm will be addressed separately.

### 2.3. Indicator Development Process

In support of the priorities and recommendations for action, the BCIPC developed a provincial working group—The Indicator Reference Group (IRG)—Consisting of experts, practitioners, and policymakers in the field of injury prevention across B.C. to support the development of injury indicators. The IRG was comprised of 17 members, including representatives from the BC Ministry of Health, the BCIPC, the B.C. Observatory for Population and Public Health, the First Nations Health Authority, the Medical Health Officers Council, and injury leads from regional Health Authorities, the BCIRPU, the B.C. Road Safety Strategy-Research & Data Committee (BCRSS-RDC), and Trauma Services B.C.

The IRG exchanged information and expertise through a series of meetings and provided expert consensus on the indicator development methodology.

The indicator development process involved a five-stage mixed methodology approach focusing on research, consultation, and collaboration with experts and the decision-makers of the provincial injury prevention programs (Figure 1), as described below.

Stage 1: Conducting an environmental scan of existing indicators of injury

A detailed international scan and review of relevant literature was performed in order to identify and categorize existing valid and evidence-informed indicators. Building on previous successful endeavours in the development of injury indicators [42,43], Medline, TRANSPORT, TRIS (Transportation Research Information Services), GEOBASE, ENVIROnetBASE, PSC, ScienceDirect, and Global Health from 2000–2017, inclusive, were searched using ‘indicator’, ‘injury’, ‘road safety’, and ‘falls’ as the main search terms. Unpublished articles, program/technical reports, government press releases, academic theses/dissertations, and conference presentations were also searched as a means of accessing relevant grey literature. Personal communication with experts and secondary searches based on referenced material in reviewed articles were also undertaken.

Stage 2: Generating expert consensus

The list of indicators from the environmental scan was presented to the IRG, who proposed further review prior to decision-maker ranking. As there was a much larger number of indicators related to road safety, an expert panel (*n* = 11) consensus meeting specific to road safety was convened to develop a shortlist of the most relevant road safety indicators for B.C., to be then ranked by decision-makers. This panel rated (as high, moderate, low, none) and ranked road safety indicators based on a previously validated set of indicator selection criteria (importance, modifiability, acceptance, and practicality) established by the IRG (Table 1) [23,47,48], and the top road safety indicators were considered for ranking by decision-makers.

Stage 3: Selection of decision-makers and conducting a survey

British Columbian agencies with influence in injury prevention were identified and contacted, including all municipalities in B.C., health, research, government ministries, law enforcement, and NGOs. Within these organizations, key decision-makers and funders with direct influence over system design, budgets, and policy relating to injury prevention programs and initiatives at the executive and strategic levels were identified. A survey consisting of road safety, seniors falls, and all-injury indicators was designed using REDCap (Research Electronic Data Capture version 8.1) hosted by the B.C. Children’s Hospital Research Institute. This is a secure web-based software platform designed to support data capture for research studies and creates databases and projects [49,50]. Decision-makers were requested to rank each indicator that fell within their sphere of influence using a four-point Likert-type response scale (none, low, moderate, and high) in terms of the following two criteria [48,51]: importance: “Regardless of your scope of authority or expertise, how important do you feel it is to achieve the target outcome measured by the indicator (i.e., preventing road, falls-related, other injuries)?” and actionability: “How useful do you consider the indicator to be for taking action aimed at implementing injury reduction strategies?”

Stage 4: Selection of final indicators by the Indicator Reference Group

Through a series of consensus meetings, the IRG reviewed decision-maker survey responses and applied inclusion criteria and thresholds to determine the final list of indicators in each priority area. This decision was made to yield a suite of indicators that represented the leverage points which were most important and actionable upon which to intervene. For road safety indicators, this included ratings from both decision-makers and subject-matter experts, while for seniors falls and all-injury indicators, only decision-makers’ ratings were available for thresholding. Indicators were included if the percentage of decision-makers and experts who rated the indicator as high or moderate in the selection process were above an established threshold.

Stage 5: Specification of indicators

At this stage, indicators were disaggregated by demographics, including age, sex, and health authority, to provide additional insight regarding vulnerable groups, and to also allow local agencies to act upon these indicators. After the lists of road safety, seniors falls, and all-injury indicators were finalized, each indicator was grouped together with ‘like’ indicators. This was carried out, recognizing that many of the indicators were merely a disaggregation or variation of a larger metric. Each grouping of ‘like’ indicators was then specified in order to define and standardize the generation of each indicator [42,48]. An indicator specification template was adapted to better fit the suite of injury indicators and the system in B.C. (Appendix A) [48]. The IRG confirmed the adapted template as well as the completed set of specification tables.

## 3. Results

Based on the in-depth search of peer-reviewed and grey international literature, as well as examination of key websites in the field of injury (stage 1), three lists of injury indicators in the priority areas reported by key stakeholder organizations and other leading jurisdictions were developed. These indicators were then filtered by combining duplicates and removing those that did not fit the B.C. context. At stage 2, seniors falls and all-injury indicators were categorized into four domains, including health status, policy, health service coverage, and health systems [52], and were reviewed by subject matter experts in terms of comprehensiveness. The road safety indicators were organized into five domains based on the ‘safe systems’ approach: outcome indicators, safe vehicles, safe roads, safe road users, and safe speed [53,54,55,56]. At stage 3, the expert panel ranked 107 road safety indicators, and the top 72 were considered for ranking by decision-makers. In the next stage (July, November, 2018), 561 decision-makers were provided with the online survey, and 128 responded (22% response rate). In stage 4, ratings by decision-makers as well as experts were included to select the final suite of road safety indicators. For several indicators, the majority of experts and decision-makers agreed upon inclusion in the final set. Some indicators were ranked as important and actionable by the decision-makers, while experts did not rank them as moderate or high. For instance, 8 indicators (3 in the domain of ‘safe systems’ and 5 in ‘road safety’) were deemed as moderate to high regarding importance and actionability by more than 60% of decision-makers, while not being ranked as such by the experts. The lowest level of consensus was observed for ‘safe vehicles’ indicators, among experts as well as decision-makers. In order to balance and integrate the opinions of decision-makers and experts and maintain indicators in each defined domain, different thresholds (from 50% to 70% with 5% intervals) were examined to come up with the most parsimonious list of indicators in each priority area without losing any domains. Final thresholds varied from 50% consensus for seniors falls, 55% for all-injury indicators, and 60% for road safety. This ensured that the final list of indicators would measure all important domains in each priority area of the injury prevention system, based on the views of decision-makers and experts. Final indicator lists were then viewed as a suite, and to ensure that they measured the whole injury system, in this stage, three road safety and one seniors falls and one all-injury indicators were added by the IRG. A final set of 47 road safety, 18 seniors falls, and 33 all-injury indicators were identified (Appendix A). In stage 5, grouping and specification of ‘like’ indicators resulted in a final list of 23 road safety indicators, with 413 items of variations and disaggregation, 8 seniors falls indicators, with 172 items of variations and disaggregation, and 13 all-injury indicators, with 529 items of variations and disaggregation (Appendix A). The final 44 injury indicator groups are listed in Table 2.

## 4. Discussion

Accurate evaluation of injury prevention initiatives requires the identification and selection of indicators that capture the full spectrum of injury severity and are sensitive to changes in the injury system [23,42,57]. Decision-makers in B.C. previously expressed interest in having a holistic and comprehensive picture of the injury system to support their decision-making processes. In earlier research, they indicated the need for injury-related information to be linked to both determinants or causes and consequences or outcomes, for the purpose of monitoring and evaluation. They expressed a desire to have the type of information which enables them to detect any unique local concerns, trends, needs, or circumstances that ought to be acknowledged and adequately addressed [58].

In this study, decision-makers with direct influence over system design, budget, and policy relating to injury prevention programs and initiatives ranked each set of indicators on importance and actionability. By considering decision-maker ranking, instead of only expert opinion as in previous studies [32,41], we anticipate an improved performance of the injury system through trust-building, advocacy and accountability, and further encouragement for resource allocation in priority areas [25,27]. Although we deployed four criteria for indicator ranking by experts, these were condensed into two (‘importance’ and ‘actionability’) for retrieving the opinions of decision-makers, in order to increase their participation rate in the indicator selection process. A Canadian study on the development of performance indicators for public health emergency preparedness showed that importance and actionability were the most relevant criteria for the early stage of indicator development [51]. These criteria were also successfully implemented in generating a set of 34 child and youth injury indicators in Canada [48].

To reach an optimum number of key indicators for the final suite, the IRG decided to set inclusion criteria as well as thresholding. Inclusion of indicators was based on the percentage of decision-makers and experts that rated the indicator as high or moderate in the selection process. This enabled inclusion of the most essential indicators that were deemed important and actionable by the majority of raters, so as to promote feasibility and the sustainability of further implementation. Although there are not many publications on thresholding for the development of indicator suites, Khan et al., implemented an a priori cut-off for consensus; indicators which reached 70% consensus as being both important and actionable were retained after round 1 of a modified Delphi technique [51]. In our study, during the deliberation on what threshold to use, it was noted that if the same threshold was applied across the injury priority areas, too many indicators would be included in certain areas, while some would not be represented in the final suite. This would result in an incomplete picture of the whole injury system. As such, the IRG decided to apply different thresholds to different injury priority areas (ranging from 50% to 60%) to ensure the complete system was measured. An interesting finding in stage 4 was that ‘safe vehicles’ indicators were less commonly suggested for inclusion in the final suite by both experts and decision-makers. This might have been because the development of vehicle safety regulations and technical standards are under the governance of Transport Canada, a federal government organization. Thus, experts and decision-makers less commonly perceived these indicators as being ‘actionable’ or ‘modifiable’ at the provincial level.

Indicators selected through thresholding were then specified in accordance with the indicator specification template [48]. During this process, two major observations were made. First, that some of the indicators on the final list were related to others, such as number and rate of unintentional injury, or provided additional context in relation to another, such as number of intersections with red light cameras and speed cameras. Therefore, some of these indicators would be more effectively viewed as variations of a single indicator as opposed to multiple separate indicators. Second, the insight that although these indicators were aimed at the provincial level, breakdowns by demographics and region (e.g., health authority) could provide the required information for local agencies to develop appropriate interventions for target groups. Disaggregation of indicators can further facilitate identification of vulnerable subgroups and determine where inequities exist. As indicators are dynamic (i.e., reflect specific time-linked situations and contexts), disaggregated indicators provide information on potential changes in the distribution patterns of health events and reflect the impact of policies and interventions [27]. In our study, indicators were grouped and expanded accordingly, resulting in each indicator having multiple items, collectively termed variations and disaggregation. This process generated 44 grouped (‘like’) indicators in the priority areas of road safety (23), seniors falls (8), and all-injury (13) to compare performance between communities, to identify gaps and opportunities for intervention, and to monitor the effect of the injury prevention strategies implemented. It was deemed that the final, shorter list of grouped indicators would be more acceptable to decision-makers and end-users and would facilitate regular and complete reporting.

Our study had some limitations, as it was an action-oriented research process, incorporating opinions of injury experts from various disciplines with standpoints of decision-makers from different sectors. While efforts were made to be objective in the process, the results depend upon the opinions of the participating experts and surveyed decision-makers. As Cryer et al. have emphasized [45,59], the newly proposed indicators should be subjected to a validation process before being widely promulgated. However, formal validation may prove difficult, due to the lack of a ‘gold standard’ as a basis for comparison [48]. The other potential limitation of our study is a lack of data to populate some of the indicators. However, having ‘actionability’ as one of the two main criteria for the selection of indicators by the decision-makers increases the likelihood that data are available for most indicators, or that availability is feasible upon data linkage and effective collaboration among sectors.

The potential next steps in research, policy, and practice will focus on the utilisation of existing data for populating the indicators as well as promoting advocacy for the creation of de novo databases, where required. Maintaining the dialogue with injury decision-makers is also essential to seek their preferred approach for communicating the indicators, as well as the appropriate platform and interval. These steps will aim to promote the utility of the suite of indicators to serve as a North Star for reliable navigation of the impact of efforts in injury prevention initiatives in priority areas. The expected impact of this process would be higher efficiency and consistency in tracking the progress toward decreasing the societal burden of injury.

Recommendations for future research include the implementation of robust methodologies (including modified Delphi and participatory action research) for integration of expert opinions and decision-maker perspectives in other injury prevention priority areas. Exploring the obstacles that limit sustained partnership between experts and decision-makers in different jurisdictions and contexts would also be an invaluable and essential step. Recognition of these barriers (including the privileging of scientific evidence over decision-makers’ experience and knowledge, poor communication or lack of adequate trust in the past, differing priorities and reward structures, limited organizational capacity or available resources [21]) can be followed by effective interventions to minimize the challenges and decrease the evidence-practice gap. Longitudinal studies can also be beneficial to objectively track the impact of indicator implementation in cost-effective resource allocation and injury control.

## 5. Conclusions

This paper addresses a knowledge gap in developing a suite of injury indicators that aligns with expert and decision-maker opinions and enables policy-relevant whole-system reporting. In this study, implementation of a five-stage mixed methodology approach focused on research and consultation and collaboration with experts and decision-makers concluded in the development of 44 injury indicators in priority areas. This ‘science-practice’ integrated approach was deemed essential as a ‘systemic’ approach to injury prevention, enabling whole-system monitoring and evaluation [60].

Although some previous research endeavors have also exercised the development of injury indicators, our study was rather unique as it is among the few which succeeded in developing a ‘suite’ of indicators, and it covered more than one injury priority areas [37,61]. Our study process also benefited from both expert opinion and decision-makers’ views. Indicators that are agreed upon by decision-makers can potentially bring about greater success in sustained injury prevention initiatives [25]. This undertaking was based on the idea that injury control is the ‘shared responsibility’ of experts and decision-makers, and through partnership, each can contribute by providing diverse and valuable perspectives to the co-creation of the discourse [25,62]. Furthermore, the IRG evaluated each of the indicators after achieving consensus and thresholding and had the opportunity to suggest additional indicators for inclusion in the final suite.

Our suite of indicators counts as a useful addition to the injury prevention methodology literature in that it represents important and actionable aspects of the injury system. While being specific to the B.C. context, our suite of indicators can contribute to national and global efforts to gauge progress in reducing injury burden, given the indicators were derived based on an in-depth international search of existing injury indicators. The selected priority areas were not only the main injury priorities in B.C., but were among the top 25 leading causes of disability-adjusted life-years (DALYs) throughout the world [63]. This illuminates the way forward for other jurisdictions to validate and customize our suite of indicators for their context. These proposed indicators depend upon meaningful and systematic injury surveillance, adequate coverage, and reporting, so as to guide and evaluate injury prevention initiatives.

## Figures and Tables

**Figure 1 ijerph-18-11837-f001:**
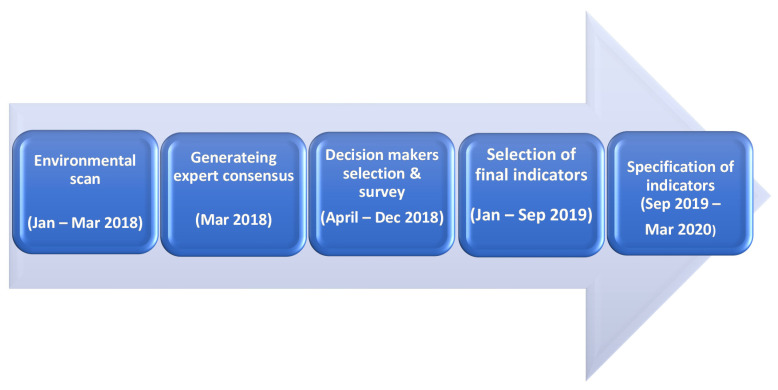
Stages of developing policy-relevant indicators for injury prevention in British Columbia.

**Table 1 ijerph-18-11837-t001:** BC injury indicator selection criteria used for road safety expert consensus.

Criteria	Description
Importance	The indicator measures an injury issue that is large; is trending; has a major life impact affecting a B.C. population; is urgent; and has ethical/social justice/equity impacts
Modifiability	The indicator will measure evidence-informed injury interventions or sets of evidence-informed injury prevention interventions and the extent to which we are changing the conditions that contribute to final health and safety outcomes
Acceptance	The indicator measures an evidence-informed injury prevention intervention supported by and understood by public, government municipality, and other relevant stakeholders
Practicality	The indicator measures an intervention (current or potential) that is practical and feasible to implement in the B.C. context

**Table 2 ijerph-18-11837-t002:** Final list of grouped (‘like’) injury indicators (23 Road Safety, 8 Seniors Falls, and 13 All-Injury).

Grouped (‘Like’) Indicators	Included Indicators *
1. Access to timely and appropriate care	O1,2
2. Road-related fatalities	O3-5
3. Road-related serious injuries	O6,7
4. Road-related ED visits	O11
5. Automated Speed enforcement	SS1,2; SRU2-4
6. Speed compliance	SS5-8
7. Signalised intersection safety	SRU5,6
8. Designated heavy truck traffic route	SR8
9. Pedestrian safety in school zones	SR1
10. Pedestrian safety	SR2, SR3
11. Enhanced road designs	SR4, SR5
12. Passive safety technology	SV1
13. Traffic enforcement	SRU1
14. Traffic legislative initiatives to enhance traffic safety	SRU7-9
15. Traffic violations	SRU10-14
16. Safe and unsafe driving behaviours	SRU15-17
17. Active transportation	SRU18,19
18. Unsafe speed and traffic crashes	O8-10
19. Speeding related vehicle impoundments	SS3
20. Evidence-based speed limit approach	SS4
21. Unprotected and protected on-road bike lanes	SR6
22. Multi-use pathway	SR7
23. Road-related injury costs	HS18-20 ^#^
1. Fall-related fatalities	HS1
2. Fall-related injury hospitalization	HS2,3
3. Fall-related ED visits and repeat visits	HS4,5
4. Fall-related costs	HS6-8
5. Wait time for surgery	HS9
6. Health service coverage	HSC1-3
7. Fall prevention designated staff	PLC1
8. Availability of fall prevention resources and plans	PLC2-6
9. Number and rate of unintentional injury fatalities	HS1-3
10. Number and rate of unintentional injury hospitalizations	HS4,5; HS10-12
11. ED visits for unintentional injuries	HS13-15
12. PYLL, DALY, and cost of injury	HS17-20
13. Poisoning helpline utilization	AI11
1. Self-reported unintentional injuries	HS16
2. Treatment coverage for substance use	HS21
3. Percentage of adult binge drinking	AI1
4. Communities with access to water safety programs	AI3,4
5. Injury prevention legislation and policy 1	AI5-8
6. Injury prevention legislation and policy 2	AI9
7. Percentage of bicycle helmet use	AI10
8. Availability of fire and ambulance services	AI12

* SS: Safe Speed; SR: Safe Road; SV: Safe Vehicle; SRU: Safe Road User; O: Outcome; HS: Health Systems; HSC: Health Service Coverage; PLC: Policy; AI: Additional Indicators. (The full indicator list and names appear in Appendix A). ^#^ From ‘All-Injury’ priority area.

## Data Availability

The aggregated data presented in this study are available on request from the corresponding author.

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
