# Peer review of "Development of Policy-Relevant Indicators for Injury Prevention in British Columbia by the Key Decision-Makers"

_ijerph, 2021, doi:10.3390/ijerph182211837_

Round 1
Reviewer 1 Report
In the abstract of the article, it is not clear the methodology used by the authors throughout their research. It would be very important that the authors include this information in the abstract
[line 64-65] The authors state in the article: “Indicators are, in fact, summary measures that can shed light on a situation 64 which is not obvious when considered by itself [15].” - It would be important, and beneficial to the scientific quality of the article, if the authors referred to some of the most important indicators, to help readers...
It would be important if the article had identified the "research question"! In other words, the purpose of this research, which led to the publication of this article.
[line 187-188] The authors state in the article: “Based on the in-depth search of peer-reviewed and grey literature, as well as examination of key websites in the field of injury…” - But which databases were consulted? What criteria were used to define the databases consulted? What were the "key words" used for the identification of the articles selected for the identification of the indicators? Many doubts that need to be clarified by the authors so that the article can have the desired scientific level...
The authors left no recommendations for future research. It would be very important that the authors, given the knowledge and experience acquired with this research and publication, recommend future work/research so that the scientific community could continue with this research. I believe that this work has great potential...
Author Response
"Please see the attachment."

Reviewer 2 Report
- This manuscript describes the development of injury prevention indicators, with a focus on decision makers. This is not a traditional research study, but rather a useful description of the process they used to get to their final indicators.
- The authors do a very nice job in the background describing what "indicators" are, and how they are relevant to the injury prevention field, and British Columbia.
- Line 106 - it would be helpful if more information was provided on what their "modified Delphi approach" was. who participated in this? how was different from a normal Delphi approach?
- Line 117 - the IRG seems like a very important group throughout this entire development process. Readers would benefit from learning more about them. How many in total were on the IRB? Subject matter experts in what? Does policy-makers mean government representatives? If so, from where? what areas did the "practitioners" work in?
- Line 134 - add quotes around the search terms used.
- Line 148-162 - Stage 3 - how people participated in this stage? They ranked each indicator on Importance and Actionability. Please explain why the criteria used Stage 2 were not also used here. It seems to me that Modifiability and Acceptance would still be important.
- Line 167 - what "inclusion criteria" were used? and how were the "thresholds" established?
- Line 197 - why the top 72? 72 seems like an arbitrary number.
- Line 215 - 47, 18, and 33 also seem like arbitrary numbers. How were the final numbers determined?
- Line 222 - Table 2 - I see in appendix 2 Total road-related fatalities per 100,000 population is indicator number 40, but it seems conspicuous that "fatalities" isn't it's own indicator in table 2 for road safety. Fatalities is the first one listed for both balls and all-injury. Additionally, for All-Injury, numbers 1 and 2 say "number and rate", where as this is not specified for Road Safety or Senior Falls. I see these are listed out in the Appendices, but if someone is just reading the table, the indicators listed don't seem to accurately communicate what these are. I personally think rates are always better indicators than absolute numbers, but that is just my opinion.
- I very much like how decision makers were engaged in this process.
- As a reader, I feel like a missing piece of this story is the plan for how these indicators are going to be used beyond a generic statement about being used to inform decision makers. The final indicators in all domains make sense to me, but it seems like it would be an incredible amount of work to get access to data sources needed (are data even available for all the indicators?), then analyze those data and synthesize and communicate findings into something that is this meaningful and impactful? Who is responsible for doing all this? If there is no plan, then this seems like an academic exercise and less interesting. How all this becomes action is what's most important.
Author Response
"Please see the attachment."

Reviewer 3 Report
I carefully reviewed your manuscript. The study is interesting and original.
I approve the publication of this paper after major revision.
General comments:
Insert the limitations of the study
Introduction
Please change background with introduction
Please: Write the background more concisely
Table
In the table 1: Please, put the text of the second column aligned to the left
Author Response
"Please see the attachment."

Round 2
Reviewer 1 Report
The article lacks a strong literature review, so the authors should review and improve this.
Authors should review the formatting of the References at the end. Why do they put "Available online"? It is not expected that the scientific paper has so many references, based on information on web pages. It is recommended that the authors review and improve the literature review, because the literature review should be done based on scientific articles, preferably in indexed journals (ISI+SCOPUS), so that this research is based on validated scientific knowledge. As it currently stands, this situation jeopardises all research, given the lack of a true scientific basis.
Why did the authors focus only on Canadian indicators in the Literature Review, and not look at what is happening in other countries? This is supposed to be a scientific paper, not an academic paper! It is important that the authors review this situation and research scientific papers, which analyse and address this situation. Very important!
The authors have rightly identified the research question that supports and justifies the authors' research. However, it was expected that the conclusions would clearly and objectively answer this research question, which is not the case. It would be very important if the authors could improve this situation.
It would be very important for the authors to recommend future works so that the scientific community can continue the authors' research. It does not make sense for this research to stop here. Why not give it continuity? This would be a great contribution that the authors could give to the scientific community, in the search for knowledge to improve the welfare of society! It is recommended that the authors recommend future works, to give continuity to this research!
Reviewer 3 Report
Dear Authors,
I carefully reviewed your manuscript.
I appreciate changes you have made in the manuscript.
I think that it could be interesting for our readers in its current form.
After revision the paper has consistently improved in quality.
I approve of the submission.
Author Response
We convey our appreciation to the reviewer for their careful review of the manuscript and the positive comment.